# Thrombotic Thrombocytopenic Purpura: Pathophysiology, Diagnosis, and Management

**DOI:** 10.3390/jcm10030536

**Published:** 2021-02-02

**Authors:** Senthil Sukumar, Bernhard Lämmle, Spero R. Cataland

**Affiliations:** 1Division of Hematology, Department of Medicine, The Ohio State University, Columbus, OH 43210, USA; senthil.sukumar@osumc.edu; 2Department of Hematology and Central Hematology Laboratory, Inselspital, Bern University Hospital, University of Bern, CH 3010 Bern, Switzerland; bernhard.laemmle@uni-mainz.de; 3Center for Thrombosis and Hemostasis, University Medical Center, Johannes Gutenberg University, 55131 Mainz, Germany; 4Haemostasis Research Unit, University College London, London WC1E 6BT, UK

**Keywords:** thrombotic thrombocytopenic purpura, TTP, ADAMTS13, treatment, diagnosis, follow-up, review, caplacizumab

## Abstract

Thrombotic thrombocytopenic purpura (TTP) is a rare thrombotic microangiopathy characterized by microangiopathic hemolytic anemia, severe thrombocytopenia, and ischemic end organ injury due to microvascular platelet-rich thrombi. TTP results from a severe deficiency of the specific von Willebrand factor (VWF)-cleaving protease, ADAMTS13 (a disintegrin and metalloprotease with thrombospondin type 1 repeats, member 13). ADAMTS13 deficiency is most commonly acquired due to anti-ADAMTS13 autoantibodies. It can also be inherited in the congenital form as a result of biallelic mutations in the *ADAMTS13* gene. In adults, the condition is most often immune-mediated (iTTP) whereas congenital TTP (cTTP) is often detected in childhood or during pregnancy. iTTP occurs more often in women and is potentially lethal without prompt recognition and treatment. Front-line therapy includes daily plasma exchange with fresh frozen plasma replacement and immunosuppression with corticosteroids. Immunosuppression targeting ADAMTS13 autoantibodies with the humanized anti-CD20 monoclonal antibody rituximab is frequently added to the initial therapy. If available, anti-VWF therapy with caplacizumab is also added to the front-line setting. While it is hypothesized that refractory TTP will be less common in the era of caplacizumab, in relapsed or refractory cases cyclosporine A, *N*-acetylcysteine, bortezomib, cyclophosphamide, vincristine, or splenectomy can be considered. Novel agents, such as recombinant ADAMTS13, are also currently under investigation and show promise for the treatment of TTP. Long-term follow-up after the acute episode is critical to monitor for relapse and to diagnose and manage chronic sequelae of this disease.

## 1. Introduction

### 1.1. History of Thrombotic Thrombocytopenic Purpura

In 1924, Dr. Eli Moschcowitz described a previously healthy 16-year-old girl who became acutely ill with fever, weakness, focal neurological symptoms, and severe thrombocytopenia. Ultimately, she became comatose and died after one week. Autopsy revealed widely disseminated thrombi in the terminal arterioles and capillaries of various organs but the underlying etiology of this mysterious illness was unknown [1,2]. This poorly understood condition was named thrombotic thrombocytopenic purpura (TTP) by Singer in 1947 [3]. Two decades later, Amorosi and Ultmann introduced the classic diagnostic pentad of TTP consisting of fever, thrombocytopenia, hemolytic anemia, renal injury, and neurological manifestations. Their case series and review of the literature also highlighted the >90% mortality rate of this devastating condition [4]. Shortly thereafter, case reports detailing the successful treatment of congenital TTP (cTTP) patients with infusions of plasma led to the conclusion that a deficiency of an unknown plasma factor contributed to the disease [5,6]. In 1982, Moake et al. first identified “unusually large” von Willebrand factor (VWF) multimers in the plasma of four chronic relapsing TTP patients—similar to the large multimers synthesized and secreted by human endothelial cells in culture. They hypothesized that these hyperadhesive ultralarge VWF (ULVWF) multimers were due to a suspected deficiency of a VWF depolymerase present in normal plasma [7]. Their hypothesis was reinforced when a highly effective therapy for TTP, plasma exchange, was described in 1991. The treatment of immune-mediated TTP (iTTP) was revolutionized and the mortality rate was improved from >90% to 10–20% with prompt therapy [8]. Five years later, a novel metalloprotease which specifically cleaved ULVWF was purified from human plasma [9,10]. A severe deficiency of this protease was noted in TTP patients, both through acquired autoantibodies and through an inherited deficiency [11,12]. In 2001, this was subsequently identified as ADAMTS13 (a disintegrin and metalloprotease with thrombospondin type 1 motifs, member 13), the only known function of which is to cleave VWF [13,14,15,16,17]. As of 2020, the improved molecular understanding of TTP along with study of survivors have allowed for marked advancements in diagnosis [18], treatment [19,20,21,22], and the long-term management [23,24,25] of these patients.

### 1.2. Definitions and Terminology

Thrombotic microangiopathy (TMA) is a broad term which has both pathologic (occlusive microvascular or macrovascular disease commonly with intraluminal thrombus formation) and clinical (microangiopathic hemolytic anemia (MAHA) with thrombocytopenia) definitions [26,27]. The different entities presenting with TMA findings have historically been difficult to distinguish from one another, but elucidating the pathophysiology of TTP has allowed for more accurate differentiation. As a result, standard definitions and terminology have been adopted [27,28].

TTP is characterized by MAHA with severe thrombocytopenia and variable organ ischemia, most commonly neurologic, cardiac, or renal [3,4,23,29]. The diagnosis is confirmed by a severe deficiency (<10%) of ADAMTS13 activity [11,12,27]. TTP is further divided into two categories based on the mechanism of ADAMTS13 deficiency: congenital (inherited) vs. immune-mediated (acquired). Congenital TTP, also known as Upshaw–Schulman syndrome or hereditary TTP, is defined by a persistent severe deficiency (<10%) in ADAMTS13 caused by biallelic pathogenic mutations in the *ADAMTS13* gene [27]. Immune-mediated TTP, sometimes referred to as acquired TTP, is caused by ADAMTS13 deficiency mediated by autoantibodies [12,27]. iTTP is further subdivided into primary iTTP, when there is no obvious associated disorder, and secondary iTTP, when an associated condition can be identified [27].

## 2. Epidemiology

iTTP typically presents in adulthood, accounting for 90% of cases [29]. The annual incidence is 1.5–6 cases per million per year in adults [29,30,31,32]. Discrepancies in annual incidence rate are likely due to demographic factors in the country of origin. In France and Germany, which are predominantly Caucasian, the incidence is ~1.5 cases per million per year [29,32]. The annual incidence in the U.S. is 2.99 cases per million per year, possibly a result of the higher proportion of African Americans, who have an approximately eightfold-increased incidence rate of TTP [31,33]. In a regional UK registry, the incidence rate was found to be six per million, though this could represent an overestimation as TTP was diagnosed clinically and did not rely on ADAMTS13 measurement in all cases [30].

Childhood-onset iTTP is considerably less common, comprising approximately 10% of all cases [34]. There is a scarcity of data regarding the incidence and prevalence of child and adolescent onset iTTP. The French National TMA Registry estimates the yearly incidence of childhood-onset iTTP to be 0.2 new cases per million with a prevalence of 1 case per million as of December 2015 [34]. This is consistent with the childhood iTTP incidence rate found in the Oklahoma (U.S.) registry of 0.1 cases per million [31].

Women are two to three times more likely to develop iTTP, which is consistent across registries globally [29,30,31,32,34,35,36,37]. ADAMTS13 deficiency is caused by an acquired autoimmune mechanism for the vast majority of TTP cases.

An inherited deficiency of ADAMTS13 due to mutations in the *ADAMTS13* gene occurs in approximately 3–5% of patients with TTP [29,30,31,36]. The exact prevalence of cTTP is uncertain, though some experts estimate this to be 0.5–2 cases per million; further investigation is needed [38]. cTTP often presents in childhood prior to 10 years of age [39,40,41,42] but large registries have reported that 10% of cases occur after the age of 40 [40,41,42]. cTTP accounts for a significant proportion of TTP cases in children and obstetrical TTP patients, consisting of 33% and 34% of all cases in those cohorts respectively [29,34].

## 3. Pathophysiology

### 3.1. Role of ADAMTS13 and VWF in TTP

ADAMTS13 is a critically important enzyme, synthesized in hepatic stellate cells [43,44], whose only known function is to regulate VWF multimers [9,10]. In physiologic conditions, ADAMTS13 is in a latent, closed conformation and VWF, secreted by platelets and endothelial cells, is in a globular state (Figure 1a) [45,46]. Proteolytic activity of ADAMTS13 on VWF is dependent on the conformational change of both proteins [45,46,47,48,49,50]. Under shear forces VWF unravels and exposes its A1 domain allowing for interaction with platelets through the GpIb/IX/V complex (Figure 1b) [51,52,53]. In this unraveled state, the A2 domain of VWF is elongated and exposes the ADAMTS13 binding sites [48,50] and the cleavage site Tyr1605-Met1606 [9,10]. Initial interaction of CUB1-2 domains with VWF D4-CK domains allosterically activates ADAMTS13, inducing an open conformation (Figure 1c) [47,49]. Sequential exosite interactions and binding of the disintegrin-like domain of ADAMTS13 to VWF induces further allosteric activation of the metalloprotease domain which results in proteolysis (Figure 1d) [54]. When severe ADAMTS13 deficiency (<10%) is present, ULVWF multimers can accumulate leading to unregulated platelet adhesion and aggregation, resulting in TTP with disseminated microthrombi and organ ischemia [4,7,26].

Though a severe ADAMTS13 deficiency is necessary for the development of TTP, enzyme deficiency alone may not be sufficient to induce the clinical syndrome [40,55,56,57,58]. Activation of the complement system has also been suggested to play a role in acute TTP [59,60,61,62]. In fact, ULVWF multimers serve as a scaffold for the assembly and activation of the alternative pathway of the complement system [61]. VWF acts as a cofactor for complement factor I mediated cleavage and inactivation of complement C3b, thereby regulating alternative pathway activation. This regulatory process is dependent on VWF multimer size with the smaller, physiologic VWF multimers enhancing cleavage of C3b and the ULVWF multimers losing this function [62]. Further studies have demonstrated a correlation between the presence of ULVWF multimers and higher levels of sC5b-9, C3a, and C5a [63]. Experimental mouse models have recently demonstrated a synergistic effect of ADAMTS13 deficiency and complement dysregulation. Mice with Adamts13^−/−^ or heterozygous complement factor H (CFH) hyperfunctional mutation (cfh^W/R^) alone remained asymptomatic. However, mice that were both Adamts13^−/−^ and cfh^W/R^ went on to develop clinical TMA findings [64]. Clinically, complement activation has also been reported to be associated with increased mortality from an acute TTP episode [60]. These and other findings have led to a “second hit” hypothesis, suggesting that another stressor in conjunction with severe deficiency of ADAMTS13 activity is usually required to develop clinical TTP [65,66].

### 3.2. Congenital ADAMTS13 Deficiency

cTTP (also known as Upshaw–Schulman syndrome OMIM 274150) is an autosomal recessive condition caused by biallelic mutations in the *ADAMTS13* gene located on chromosome 9q34 [14]. Approximately 200 causative mutations have been identified in more than 150 patients, which span the entire *ADAMTS13* gene [14,39,40,41,42,67,68]. The majority of ADAMTS13 mutations are confined to single families [40,42]. Missense mutations are most common (59%), followed by nonsense mutations (13%), deletions (13%), splice site mutations (9%), and insertions (6%) [67]. There is some geographic variability and certain mutations have increased frequency in different regions. Two mutations in particular, p.R1060W [39,41,42,68,69,70,71,72] and insertion c.4143_4144dupA [42,68,69,73] are more prominent in cTTP patients with European ancestry. The p.R1060W mutation, a single nucleotide variant located on exon 24, also occurred in a high proportion (75–80%) of cTTP patients that presented during pregnancy in the French and UK cohorts [70,71]. Though no definite genotype–phenotype relationships have been established [41,67], earlier onset of disease appears to be related to earlier sequence mutations in the prespacer region of ADAMTS13 [41,72,74]. Often, mutations in *ADAMTS13* result in secretion deficiencies but they can also affect ADAMTS13 activity [67,74,75,76]. Indeed, in an effort to explain the variance of clinical phenotype, residual ADAMTS13 activity of different genotypes was measured and the results showed that residual ADAMTS13 activity <3% was correlated with earlier age of disease onset, need for prophylactic plasma infusions, and an annual event rate >1 [42,74]. However, this does not fully explain the phenotypic differences in cTTP as studies have demonstrated that many patients homozygous for the c.4143_4144dupA mutation had ADAMTS13 activity <1% but widely varying clinical courses [42,69,73].

### 3.3. Acquired ADAMTS13 Deficiency

#### 3.3.1. Risk Factors

iTTP is due to acquired anti-ADAMTS13 autoantibodies [11,12]. Certain factors, such as African ancestry and female sex, predispose to the development of these antibodies [29,30,31,32,33]. Human leukocyte antigen (HLA)-DRB1*11 and HLA-DQB*03:01 alleles are also overrepresented in white iTTP patients, with HLA-DRB1*04 having a protective effect [77,78,79,80]. The frequency of the HLA-DRB1*04 allele is dramatically decreased in iTTP patients with African ancestry, indicating that a low natural frequency of this allele may contribute to the greater risk in this population. However, there does not appear to be an increased risk of mortality in these patients [33]. An analysis of Japanese patients identified HLA-DRB1*08:03, HLA-DRB3/4/5*blank, HLA-DQA1*01:03, and HLA-DQB1*06:01 as predisposing factors for iTTP, with HLA-DRB1*15:01 and HLA-DRB5*01:01 being identified as weakly protective [81]. In contrast to white iTTP patients, HLA-DRB1*11 and HLA-DRB1*04 were not associated with iTTP in the Japanese [81].

#### 3.3.2. Anti-ADAMTS13 Autoantibodies

Anti-ADAMTS13 autoantibodies are largely divided into two categories: inhibitory and non-inhibitory. Inhibitory antibodies neutralize the proteolytic activity of ADAMTS13 and non-inhibitory antibodies bind to the protease, accelerating its clearance from plasma [11,12,82,83,84]. It was previously widely held that inhibitory antibodies were the main cause of ADAMTS13 deficiency, but recent studies have demonstrated that antigen depletion also significantly contributes to deficiency [85]. Even a small amount of anti-ADAMTS13 autoantibodies can induce ADAMTS13 deficiency [86]. Anti-ADAMTS13 autoantibodies have been found against all domains of ADAMTS13, indicating a polyclonal immune response. However, the spacer domain of ADAMTS13 has been identified as an immunogenic region, as anti-spacer antibodies are present in most iTTP patients [85,86,87,88,89,90,91,92]. Recently, anti-ADAMTS13 autoantibodies that induce the open conformation of ADAMTS13 have been identified [18,93]. The role these conformation-changing antibodies play in the pathophysiology of TTP and their clinical significance is still being explored.

The most common isotype class of anti-ADAMT13 autoantibodies are IgG, followed by IgA and IgM (20% of cases). Among the IgG isotype, the IgG4 subclass is most common, followed by IgG1 [83,91,94,95,96,97,98]. During acute episodes of iTTP, approximately 75% of cases have detectable free anti-ADAMTS13 IgG [29]. The anti-ADAMTS13 autoantibody isotype may contribute to the severity of the disease phenotype. High IgA antibody titers were suggested to be associated with lower platelet counts, increased mortality, and a worse prognosis [94,95,96]. Though no bacterial or viral infections are known to directly lead to iTTP, molecular mimicry between ADAMTS13 and certain pathogens such as influenza A [99], *Helicobacter pylori* [100], *Legionella* [101], hepatitis C virus [102], and HIV [103] may evoke an immune response [91,104].

#### 3.3.3. Immune Complexes

In addition to free anti-ADAMTS13 autoantibodies, immune complexes containing ADAMTS13 have also been found in 39–93% of patients during acute iTTP [105,106,107]. Given that C3a and C5a are elevated during the acute iTTP episode, this could suggest that the complement is activated through the classic pathway, via ADAMTS13 antigen-antibody immune complexes; the elevated levels of factor Bb, however, suggest activation of the alternative pathway [60,63,104,105,108]. The clinical significance of complement activation in TTP is still unclear, though it further supports the “second hit” hypothesis that another physiologic stressor in conjunction with severe ADAMTS13 deficiency is required to induce the clinical syndrome [65,66].

#### 3.3.4. Primary and Secondary iTTP

iTTP is classified as primary when no obvious underlying associated disease can be determined and as secondary when a defined underlying disorder is identified [27]. The majority of iTTP cases are primary. Secondary iTTP can be associated with infections as mentioned previously, though the best evidence is its association with HIV [103,109,110]. Acute stressors, such as pancreatitis, may induce secondary iTTP [111]. Many drugs have also been implicated in secondary TMA but are only rarely accompanied by ADAMTS13 deficiency, indicating that they mostly represent a separate drug-induced TMA (DI-TMA) and not TTP [112]. One exception is ticlopidine, which has been associated with severely deficient ADAMTS13 and this condition may be considered as secondary iTTP [113]. Notably, not all thienopyridine-derivatives (ticlopidine, clopidogrel, and prasugrel) are associated with TTP. Of 97 cases of TMA associated with ticlopidine, 80% had severely deficient ADAMT13 activity confirming the diagnosis of TTP. A clear causal relationship, however, has not been confirmed between the use of ticlopidine and the development of anti-ADAMTS13 antibodies. In 197 patients with clopidogrel associated TMA, 0% had severely deficient ADAMTS13 [114], which is consistent with DI-TMA, not TTP. Secondary iTTP can also be associated with various autoimmune conditions, though it is most commonly associated with systemic lupus erythematosus (SLE) [23,24,115,116,117]. In either primary or secondary iTTP, prompt therapy is essential. Secondary iTTP typically also requires treatment of the underlying condition in addition to standard TTP therapies.

## 4. Diagnosis

### 4.1. Clinical Presentation

Previously, TTP was defined by a clinical “pentad” consisting of fever, microangiopathic hemolytic anemia, thrombocytopenia, neurological deficits, and renal insufficiency [4]. However, the pentad was reported at a time before the effectiveness of plasma-based therapy in treating TTP was firmly established. Today, the presence of thrombocytopenia and MAHA alone, without an alternative explanation, should prompt serious consideration of the diagnosis of TTP or another TMA. Large cohort studies from various registries worldwide indicate that less than 10% of patients with acute TTP present with all five symptoms [29,30,31,35,36,37]. In fact, the clinical features of acute TTP can be extraordinarily diverse and a high degree of suspicion is required to diagnose TTP and promptly initiate appropriate management [118]. The differential diagnosis for patients with possible TTP is broad and described in Table 1. In obstetric patients with TMA, hemolysis, elevated liver enzyme, and low platelet (HELLP) syndrome and preeclampsia should be ruled out prior to evaluating for other conditions such as iTTP, cTTP, or complement-mediated hemolytic-uremic syndrome (CM-HUS) [27,112,119].

Acute TTP almost uniformly presents with severe thrombocytopenia (typically <30 × 10^9^/L) and microangiopathic hemolytic anemia, often with evidence of erythrocyte fragmentation on the peripheral blood smear [119]. Frequently, other classical parameters of hemolysis are also present, including an undetectable haptoglobin concentration accompanied by an elevated reticulocyte count, elevated total bilirubin (predominantly unconjugated), and an elevated lactate dehydrogenase (LDH) level, a marker for both red cell destruction and organ ischemia [120]. Coombs’ testing is usually negative and coagulation parameters are not severely deranged in TTP.

Signs and symptoms of organ ischemia due to microthrombi formation are variable at presentation. More than 60% of patients have neurological manifestations which range broadly from mild confusion or altered sensorium to stroke, seizures, or coma [25,29,30,36,37]. Gastrointestinal ischemia is present in 35% of patients and can result in abdominal pain, nausea, and diarrhea [29]. Evidence of myocardial ischemia is present in a quarter of acute TTP patients and can be characterized by an abnormal electrocardiogram, or more commonly, elevated cardiac troponin-I measurements. Cardiac symptoms consistent with congestive heart failure or myocardial infarction can also be seen [121]. Renal injury is not uncommon in TTP, though acute renal failure requiring renal replacement therapy is quite rare in iTTP. Hematuria and proteinuria are the most commonly seen renal manifestations. Though modest renal insufficiency may occur, most patients present with a creatinine below 2 mg/dL [122,123,124,125]. Severely deficient ADAMTS13 activity serves to confirm the diagnosis of TTP [11,12,27].

### 4.2. ADAMTS13 Investigation

#### 4.2.1. ADAMTS13 Activity

Assaying the ADAMTS13 activity is the first test which should be undertaken in patients with a suspected TMA. Severe ADAMTS13 deficiency, which is defined by an activity level <10%, is required to confirm the diagnosis of TTP (Figure 2) [119]. ADAMTS13 activity assays are based on degradation of either full-length VWF or synthetic peptides of VWF by ADAMTS13 in the plasma sample being tested. VWF cleavage products are detected by fluorescence resonance energy transfer (FRETS), enzyme-linked immunosorbent assays (ELISAs), surface-enhanced laser desorption/ionization time-of-flight (SELDI-TOF)-mass spectrometry, electrophoresis, reduced collagen binding, or reduced ristocetin-induced platelet agglutination [11,12,126,127,128,129,130,131,132,133]. Though multiple assays have been developed, the FRETS-VWF73-based assay [128,134] is most commonly used in clinical settings [135] and is considered as the reference method for ADAMTS13 activity, typically calibrated against the World Health Organization International Standard ADAMTS13 plasma (normal 100%) [136]. However, ADAMTS13 activity testing is labor intensive, time consuming, and limited to reference laboratories typically. Though the FRETS assay can be completed quickly, the turnaround time for results can be three to six days as it is typically performed only in reference centers. Given the variability in ADAMTS13 testing turnaround time for any individual center, point-based scoring systems which predict the probability of severely deficient ADAMTS13 have been developed to avoid delays in prompt treatment initiation [122,137,138]. Importantly, these scores are not meant to replace ADAMTS13 testing but to aid decision making until test results are available. Recently, fully automated chemiluminescence immunoassays have been developed with drastically reduced analytical times of approximately 30 min [139,140]. In addition, a semiquantitative ADAMTS13 activity assay has also been developed which provides an easily interpreted four-level indicator of ADAMTS13 activity, allowing identification of activity levels < 10% [141]. A potential advantage of such an assay is rapid screening for severely deficient ADAMTS13 activity which can be utilized at non-specialized centers to facilitate referral to tertiary centers for additional testing and management.

#### 4.2.2. Anti-ADAMTS13 Autoantibodies

When severely deficient ADAMTS13 activity is confirmed, the next step of investigation is to determine if an antibody inhibitor to ADAMTS13 is present [11,12]. Understanding the mechanism of ADAMTS13 deficiency is critical in differentiating iTTP from cTTP and has important treatment implications. This distinction is also especially important in children and obstetrical patients, owing to higher rates of cTTP in these cohorts [29,34,71]. ADAMTS13 autoantibodies, predominantly anti-ADAMTS13 IgG, can be readily detected using in-house or commercial ELISA kits by laboratories [83,94]. A Bethesda assay can only detect ADAMTS13 autoantibodies which functionally inhibit ADAMTS13 (inhibitory antibodies), unlike the anti-ADAMTS13 IgG ELISA which can detect both inhibitory and non-inhibitory antibodies [25,142]. For both inhibitory and non-inhibitory anti-ADAMTS13 autoantibodies, assays only detect free autoantibodies whereas those bound to ADAMTS13 (immune complexes) are not detected by standard assays. In patients who have persistent severe ADAMTS13 deficiency during periods of remission and in whom no inhibitory autoantibody is detected, *ADAMTS13* gene analysis should be pursued to confirm a diagnosis of cTTP [38].

#### 4.2.3. ADAMTS13 Antigen

ADAMTS13 antigen can be measured by ELISA but this is not yet part of routine clinical practice. A recent study evaluated the prognostic value of anti-ADAMTS13 autoantibody titers and antigen levels in patients with iTTP [143]. Patients in the lowest quartile, with an antigen level <1.5%, had a mortality rate of 18% compared with a mortality rate of ~4% for those in the highest quartile, with an antigen level >11%. Those in the lowest antigen quartile and the highest antibody quartile had the highest mortality rate of 27%. This suggests that there could be some prognostic value for this test and that it has the potential to be incorporated in clinical practice in the future.

### 4.3. Emerging Biomarkers

It has previously been demonstrated that ADAMTS13 circulates in the “open” conformation in iTTP patients during the acute phase [93]. Recently, anti-ADAMT13 autoantibodies were revealed to induce the open ADAMTS13 conformation. Additionally, the open ADAMTS13 conformation preceded significant decrement in ADAMTS13 activity in one patient followed longitudinally [18]. While these findings warrant further study, there are many potentially important implications with regard to treatment and long term follow-up. As discussed previously, though it is a major risk factor, not all patients who have undetectable ADAMTS13 activity in remission uniformly go on to relapse [55,56,58,144]. However, being able to identify the open versus closed conformation of ADAMTS13 may potentially be useful to decide on the necessity of prophylactic therapy in select iTTP patients during remission.

## 5. Acute Management

TTP is a clinical emergency and in patients with suspected TTP treatment should be initiated promptly as delays in therapy may result in significant morbidity and mortality. Often therapy decisions are required prior to the availability of confirmatory ADAMTS13 testing. A blood sample for ADAMTS13 activity testing should immediately be obtained from a patient with TMA and frontline therapy can then commence based on clinical presentation alone. Severe ADAMTS13 deficiency is still required to confirm the diagnosis but should not delay the initiation of treatment [145]. Below are definitions of treatment response from the International Working Group for Thrombotic Thrombocytopenic Purpura [27]:Clinical response—a normalization of the platelet count to a level greater than the lower limit of the established reference range (150 × 10^9^/L) and the LDH level to <1.5 × the upper limit of normal (ULN). If initial presentation is severe with evidence of significant end-organ damage, stabilization of these parameters with improvement in function should also be required to qualify for a clinical response.Clinical remission—a sustained clinical response which is maintained for >30 days after the cessation of plasma exchange.Exacerbation—a decreasing platelet count with rising LDH and the need to restart plasma exchange therapy within 30 days of cessation after an initial clinical response is noted.Relapse—a fall in platelet count below the lower limit of the established reference range (~150 × 10^9^/L), with or without clinical symptoms, during a clinical remission that requires reinitiating therapy. ADAMTS13 activity will most likely be <10%.Refractory TTP—persistent thrombocytopenia (platelet count <50 × 10^9^/L, without increment) and persistently elevated LDH (>1.5 × ULN) despite five plasma exchange treatments in conjunction with adequate steroid treatment. If platelet count remains <30 × 10^9^/L, this is classified as severe refractory TTP.

### 5.1. iTTP

Patients with both primary and secondary iTTP should be treated similarly in the acute inpatient setting. Importantly, patients with secondary iTTP should also have the underlying etiology managed appropriately in addition to the acute iTTP event. For example, in a patient with secondary iTTP due to underlying HIV infection, appropriate antiretroviral therapy would also be warranted in addition to management of TTP.

#### 5.1.1. Plasma Exchange

Therapeutic plasma exchange (TPE) with fresh frozen plasma (FFP) replacement is the foundation of front-line therapy for TTP [8]. The proposed mechanism of TPE is that it supplies adequate levels of ADAMTS13 while removing circulating anti-ADAMTS13 autoantibodies. Delays in therapy can lead to early mortality, which may be preventable with prompt initiation of TPE [146]. Typically 1–1.5× plasma volume exchange is performed for the first three days, followed by 1× plasma volume exchange each day thereafter [8]. While there is no optimal duration of therapy or pre-specified number of procedures required, therapy should be continued daily until clinical response is achieved and sustained for two days. In patients with refractory TTP or evidence of progressive end organ damage, more intensive therapy, such as twice daily TPE, may be considered [147]. The efficacy of this approach is difficult to determine as it is usually accompanied by the addition or intensification of concurrent therapies. Generally, there are no significant differences between readily available therapeutic plasma replacement products [148,149]. Previously, cryosupernatant plasma devoid of ULVWF multimers was suggested to be more efficacious than fresh frozen plasma [150], but equivalency of these plasma products was demonstrated in a small randomized controlled trial [151].

#### 5.1.2. Immune Suppression

In conjunction with TPE, immunosuppressive therapy is a cornerstone of acute iTTP management. The general principle of therapy is to target antibody production to allow for recovery of circulating levels of ADAMTS13. Therapy is typically started concurrently with TPE.

Glucocorticoids: steroids are widely used in conjunction with TPE at the initiation of therapy for acute iTTP. Though there are no randomized clinical trials comparing TPE with steroids vs. TPE alone, there is high biological plausibility for concurrent immunosuppression given the autoimmune nature of the condition. A small prospective randomized controlled trial comparing prednisone with cyclosporine A as an adjunct to TPE demonstrated that prednisone was superior in the initial treatment of iTTP [152]. This is also the only prospective randomized trial confirming the efficacy of steroids in the acute setting in decreasing anti-ADAMTS13 IgG and thereby increasing ADAMTS13 activity. No optimal dose or route of administration has been identified. High dose pulse steroids with methylprednisolone 10 mg/kg/day for three days followed by 2.5 mg/kg/day thereafter may be more efficacious than 1 mg/kg/day dosing [153]. Most standards of practice recommend oral prednisone 1 mg/kg/day or equivalent [149], gradually tapered over 3–4 weeks after clinical response is achieved. In patients with severe presentations or neurological symptoms, intravenous methylprednisolone 1 g/day for three days can be considered.

Rituximab: rituximab is a monoclonal antibody against CD20, specifically targeting B-cells. Rituximab is given most commonly during the acute phase of iTTP, typically during the first days of hospitalization or shortly thereafter. A non-randomized prospective phase 2 trial has shown its safety and efficacy in the front-line setting [154]. Additionally, this trial and many observational cohort studies suggest that rituximab given in the acute phase results in fewer relapses [20,154,155,156,157]. While a lower relapse rate did not reach statistical significance in all studies [158,159], a recent meta-analysis shows that rituximab administered during an acute iTTP episode not only lowers the relapse rate vs. control, but also reduces mortality [160]. Rituximab also appears to be effective in patients with refractory TTP or poor response to TPE [20,157,158]. The standard dosing for rituximab is 375 mg/m^2^ given weekly for a total of four doses, which is recommended for both initial iTTP episodes and the acute phase of relapsing episodes. Emerging evidence for the efficacy of low dose rituximab (100 mg–200 mg/per dose) comes from a small prospective trial [161] and retrospective studies [162] but it has not yet been widely incorporated into standard practice.

Alternative immunosuppressive therapies: in patients with contraindications to steroids or with refractory disease, cyclosporine A can be effective [19,163]. Mycophenolate mofetil has also been used with success in some case reports [164,165]. Prior to the use of rituximab, vincristine was used for refractory disease, but this is no longer preferred [166]. Bortezomib, a proteasome inhibitor targeting plasma cells, has been used successfully as an alternative agent to rituximab [167,168]. Cyclophosphamide and/or splenectomy are also options for refractory or chronically relapsing cases [169].

#### 5.1.3. Anti-VWF Strategy

Caplacizumab: caplacizumab, a humanized immunoglobulin originally from llamas, targeting the A1 domain of VWF and thereby preventing its interaction with platelets is the first medication approved specifically to treat TTP. In the recent phase 2 TITAN [21] and phase 3 HERCULES [22] trials, caplacizumab along with TPE and immunosuppression significantly reduced time to platelet count normalization and the exacerbation rate when compared with placebo. The initial dose is 10 mg given intravenously prior to the first TPE, followed by 10 mg daily and subcutaneously thereafter. Caplacizumab is well tolerated and has a good safety profile with the most common side effect being minor bleeding, which is often easily managed [170]. By blocking microvascular thrombi formation it is hypothesized that tissue ischemia can be decreased. Caplacizumab effectively blocks the end-organ damage caused by TTP; however, concomitant immunosuppression is required as the underlying deficient ADAMTS13 function is not addressed by this therapy. It is unsurprising then that exacerbations and early relapses can occur when the drug is discontinued while ADAMTS13 activity remains severely deficient. As a result, treatment is typically continued until the recovery of ADAMTS13 activity. As a novel agent, one limitation of incorporating caplacizumab into current standard practice is its high cost. At its current price level (in the United States) as of 2020, a recent analysis suggested that the addition of caplacizumab to the front line treatment for all patients with iTTP would not be cost-effective [171]. As caplacizumab is increasingly utilized, treatment response definitions may need to be revisited in the future as platelet count alone may not be an accurate measure of disease activity.

*N*-acetylcysteine: *N*-acetylcysteine (NAC) is a mucolytic approved by the Food and Drug Administration which is predominantly used to treat lung diseases. Its efficacy in TTP has been examined given that VWF multimers polymerize in a similar manner to mucins. NAC was found to degrade ULVWF multimer strings and inhibited VWF-dependent platelet aggregation and collagen binding in vitro [172,173]. NAC has been effective in some cases of severe and refractory iTTP but only a few case reports exist to date [174,175]. Animal models examining NAC have produced mixed results. NAC was able to prevent iTTP in mice but NAC administration was not successful in resolving TTP in either mice or baboons [176].

Emerging anti-VWF therapies: in 2012, ARC1779, a nucleic acid macromolecule, or aptamer, that blocks platelet binding by the A1 domain of VWF, was evaluated in TTP patients in a small trial [177]. Nine patients were recruited to the study, seven of whom received ARC1779. The study was halted for financial reasons before sufficient patients could be enrolled to ascertain the efficacy but there were no bleeding complications, despite ARC1779 suppression of VWF function in patients with severe thrombocytopenia. Development of ARC1779 has not been continued, but the safety profile from this trial encouraged the development of second generation anti-VWF aptamers. A novel DNA aptamer, TAGX-0004, showed a stronger ability to inhibit ristocetin- or botrocetin-induced platelet agglutination/aggregation than ARC1779 and a similar inhibitory effect to caplacizumab [178]. Another synthetic aptamer, BT200, has shown inhibition of human VWF in vitro and prevented arterial thrombosis in non-human primates [179]. Further studies incorporating this approach are in development.

### 5.2. cTTP

Acute episodes in patients with known cTTP can be successfully treated with plasma infusions (FFP, 10–15 mL/kg/day). Treatment is continued until clinical response is achieved [38,41,42]. In patients with a recurring cTTP phenotype, prophylactic plasma infusions may be required. Prophylactic plasma infusions have also been shown to improve chronic symptoms not related to an acute episode [39,41]. In patients who receive chronic plasma infusions, the ADAMTS13 activity half-life has been reported to be 2.5–5.4 days [180,181,182]. Consequently, ADAMTS13 activity is expected to return to baseline activity after approximately 5–10 days. Treatments are usually given every 2–3 weeks, depending on clinical symptoms, platelet counts, and patient preferences [38,41,42,181,182].

### 5.3. Emerging Therapies

Upfront therapy of TTP has seen innovative strategies in the last five years. Recombinant ADAMTS13 (BAX 930, rADAMTS13) has shown promise in a recent phase 1/2 study in cTTP patients [183]. A phase 3 clinical trial to assess the efficacy of rADAMTS13 for prophylactic and on-demand treatment of cTTP compared to plasma infusion therapy is ongoing (https://www.clinicaltrials.gov/ct2/show/study/NCT03393975). There is also evidence that rADAMTS13 may be effective in patients with iTTP, a hypothesis that is presently being prospectively studied as well (https://www.clinicaltrials.gov/ct2/show/NCT03922308) [184].

With ever growing treatment options for the acute phase of TTP, the classic treatment paradigm is constantly being re-examined. Though TPE is the cornerstone of acute therapy, there are not insignificant risks associated with the procedures required and replacement plasma products [23]. There are an increasing number of case reports detailing treatment of acute iTTP with caplacizumab and immunosuppression, without TPE, in the context of religious beliefs prohibiting blood products [185], shared decision making [186,187], and anaphylaxis to plasma [188]. As novel treatments become readily available, acute TTP management may soon enter an era without obligatory reliance on plasma exchange.

## 6. Special Populations

### 6.1. Pregnancy

TTP in the pregnant patient presents many difficulties and challenges. These patients should be managed by a multidisciplinary team typically including hematologists, high-risk obstetricians, and, occasionally, neonatologists. Prompt recognition and differentiation from preeclampsia or HELLP syndrome followed by appropriate treatment is critical, as maternal/fetal morbidity and mortality are high if unrecognized [70]. Pregnancy can trigger acute episodes in cTTP patients who have previously been asymptomatic. Approximately 25–30% of all obstetrical TTP cases were due to cTTP in some cohorts [29,34,71]. Thus, a high suspicion for cTTP is warranted in pregnant patients and appropriate diagnostic workup should be pursued if there is no evidence of an inhibitor or anti-ADAMTS13 autoantibodies. Acute management of cTTP in pregnancy includes plasma infusions but more severe cases may require TPE [38].

In pregnant patients with iTTP, the acute phase should be managed with TPE with the addition of corticosteroids if tolerated [70]. Though corticosteroids may confer some risks if given during the first trimester, these are largely outweighed by the potential benefits in this clinical context. Further immunosuppression with rituximab has not been studied in pregnant iTTP patients and its use is not standard. Routine use of caplacizumab is not recommended given the theoretical risk of fetal hemorrhage.

In remission after an episode during pregnancy, cTTP patients may require prophylactic therapy prior to and during their next pregnancy. The recently published International Society of Thrombosis and Haemostasis (ISTH) guidelines for management of TTP state that pregnant cTTP patients should receive prophylactic plasma infusions to prevent relapse [189].

In remission after any acute episode, iTTP patients who are pregnant or could become pregnant should have ADAMTS13 monitored periodically. Severely deficient ADAMTS13 activity in pregnancy appears to uniformly predict relapse of iTTP [190]. Though there is currently a lack of strong evidence, prophylactic therapy for pregnant patients with a history of iTTP and severely deficient ADAMTS13 activity in remission is suggested due to the risk of mortality to both mother and fetus associated with relapse [191]. No standard prophylactic regimen has yet been determined for this indication.

### 6.2. Jehovah’s Witnesses/Contraindication to Blood Products

Certain groups, including Jehovah’s Witnesses, may not accept exogenous blood products on the basis of religious or other beliefs. As TPE is the foundation of management of acute episodes, this presents a unique challenge in the management of these patients. Various regimens have previously been tried, including vincristine [192] and plasma exchange with albumin [193] or cryosupernatant [194] replacement. With the use of caplacizumab alongside improved immunosuppressive therapy, successful treatment without TPE has been described not only in this patient population [185] but also in other selected patients, including one with anaphylaxis to plasma [186,187,188].

## 7. Long-Term Follow-Up and Remission Management

TTP was previously thought to only be an acute illness but long-term follow-up of TTP survivors reveals many potential chronic complications and morbidity in addition to the risk of relapse [23,24,195,196,197]. Severely deficient ADAMTS13 activity (<10–20%) in remission suggests an increased risk of relapse and maintaining activity above this level appears adequate to prevent relapse [55,56,198]. Therefore, serial ADAMTS13 activity should be monitored in patients after remission. This is routinely accompanied by a chemistry panel, complete blood count, and measurement of LDH level. After resolution of an acute episode, ADAMTS13 activity can be measured monthly for 3 months, then every 3 months for 1 year, then every 6–12 months if stable. If ADAMTS13 activity consistently decreases, then more frequent monitoring may be appropriate [191]. However, ADAMTS13 activity is not a perfect predictive biomarker and not all patients with severely deficient activity go on to relapse [58]. Further studies highlighting the role of complement activation in the presence of ULVWF multimers suggest that the addition of other biomarkers may more accurately predict relapse in asymptomatic patients [63]. Emerging biomarkers such as the “open” vs. “closed” conformation of ADAMTS13 may also help to better predict which patients with severely deficient activity will ultimately progress to another episode [18,93].

In asymptomatic patients with ADAMTS13 activity persistently <10%, preemptive therapy with rituximab can effectively prevent relapse [144,160,198]. Cyclosporine has also been used for prophylaxis [199] and can be an option for patients who do not respond to rituximab. For the chronically relapsing patient, splenectomy is a viable option. Though falling out of favor with the development of improved immunomodulatory therapy, splenectomy has both a high and a durable response rate in some case series with a 10-year relapse-free survival of 70% [200]. Splenectomy is usually efficacious, with a nonresponse rate as low as 8% in some reports [201]. It has also been shown to induce durable remissions and reduce relapse rate in some of these challenging patients [200,202]. Though previously splenectomy had increased risk for adverse events, especially when used in refractory TTP [203], improvements in surgical technique have decreased complications considerably, especially when laparoscopic technique is utilized [201].

Long-term complications are prevalent in both iTTP and cTTP patients. Many adverse health sequelae are seen in TTP survivors, including higher rates of obesity, stroke, hypertension, mood disorders, cognitive impairment, and reduced quality of life [42,195,196,204,205,206]. TTP survivors also appear to have a higher all-cause mortality than reference populations [24,197]. Low-normal levels of ADAMTS13 activity have recently been implicated as a risk factor for coronary artery disease [207,208], stroke [209], and all-cause/cardiovascular mortality [210] in the general population. While the mechanism for the development of these complications is not known, reduced ADAMTS13 activity may contribute to cardiovascular risk. Further studies investigating this relationship as well as other potential mechanisms leading to the development of these chronic complications are warranted.

## 8. Conclusions/Summary

TTP is a life-threatening illness which requires prompt recognition and management given its high mortality if left untreated. Acute management and long-term follow-up are evolving as new therapies and potential biomarkers emerge. Given the rarity of this disease, TTP registries and multicenter cohort studies are critical to continue advancing the field.

## Figures and Tables

**Figure 1 jcm-10-00536-f001:**
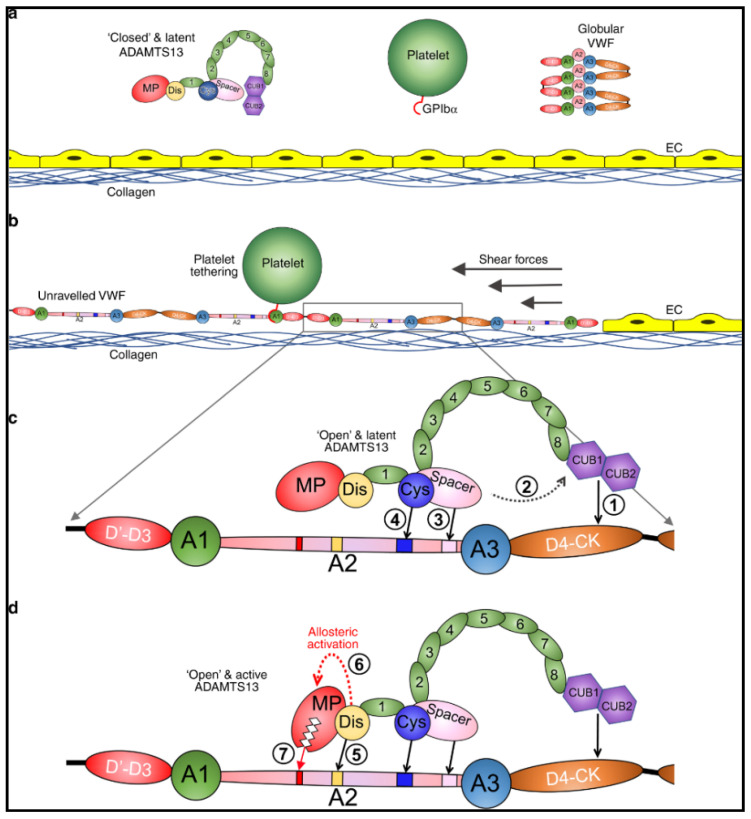
Mode of action of ADAMTS13. (**a**) Under normal circumstances, multimeric von Willebrand factor (VWF) circulates in the plasma in a globular conformation, in which its A1 domains are concealed, and so does not interact with platelets. ADAMTS13 circulates in a “closed” conformation stabilized through the interaction of the C-terminal CUB domains with the central Spacer domain. The MP domain of ADAMTS13 also has a latent conformation in which the active site cleft is occluded by the Ca2+-binding loop. This prevents ADAMTS13 from proteolyzing off-target substrates and confers resistance to plasma inhibitors. (**b**) Following vessel damage, the endothelium (EC) is disrupted to reveal subendothelial collagen. Globular VWF binds to this surface via its A3 domain and unravels into an elongated conformation in response to the shear forces exerted by the flowing blood. This reveals the A1 domain that can then capture platelets via the GPIbα receptor on the platelet surface. Unravelling of VWF also unravels the VWF A2 domain into a linear polypeptide conformation that reveals the binding sites for ADAMTS13 and the Tyr1605-Met1606 cleavage site, making it susceptible to proteolysis by ADAMTS13. (**c**) ADAMTS13 recognizes unfolded VWF through multiple interactions. (1) The CUB domains bind the VWF D4-CK domains, which (2) induces their dissociation from the Spacer domain. (3) The Spacer and (4) cysteine (Cys)-rich domain exosites recognize the C-terminal region of the unfolded A2 domain to bring the enzyme and substrate into proximity. (**d**) Once bound, (5) the disintegrin-like (Dis) domain exosite engages VWF residues Asp1614–Asp1622. This interaction (6) induces an allosteric change in the MP domain. This causes a conformational change, disrupting the “gatekeepertriad” that otherwise occludes the active site cleft, to reveal the S1′ pocket. Once allosterically activated, (7) the MP domain proteolyzes the scissile bond. Petri et al. [54], pp. 1–16. The corresponding author, James Crawley agreed to use of the Figure. No changes were made to the original figure. Creative Commons License: http://creativecommons.org/liceses/by/4.0/.

**Figure 2 jcm-10-00536-f002:**
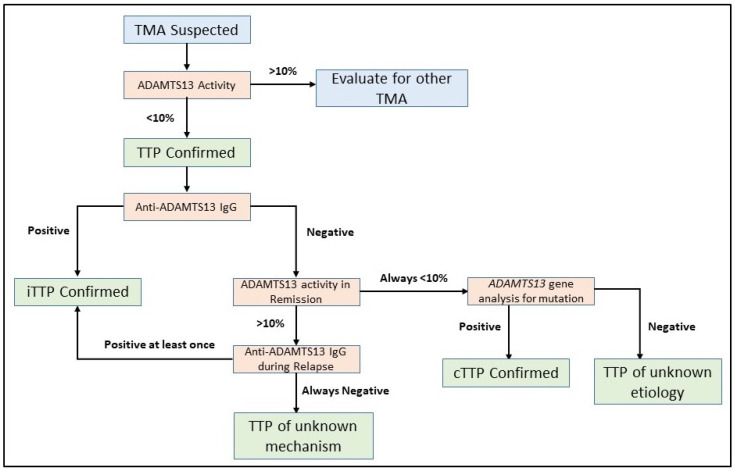
Flowchart for ADAMTS13 investigation in TTP. Severely deficient ADAMTS13 activity of <10% is required to establish a diagnosis of TTP. Further investigation of anti-ADAMTS13 IgG inhibitory autoantibodies are required to document the mechanism of ADAMTS13 deficiency. *ADAMTS13* gene analysis for biallelic mutations is reserved for selected situations to confirm a diagnosis of cTTP. In some cases, the underlying mechanism of ADAMTS13 activity deficiency is not immediately clear and repeated measurements of ADAMTS13 activity in remission and anti-ADAMTS13 IgG during relapse events can help establish a diagnosis.

**Table 1 jcm-10-00536-t001:** Differential diagnosis for patients presenting with MAHA-T [27,112].

Disease	Comment
TTP	Defined by ADAMTS13 activity <10%
IA-HUS	TMA presenting 5–7 days after infection, often hemorrhagic colitis caused by enteropathogenic *Escherichia* coli, or *Shigella.*
CM-HUS	Triggered by infections, vaccination, pregnancy, or surgeries. Diagnosis may be confirmed by complement mutations
DI-TMA	May occur with gemcitabine, bleomycin, mitomycin, quinine, cyclosporine, simvastatin, and others. VEGF inhibitors have also been implicated.
TA-TMA	May occur with hematopoietic stem cell transplantation or solid organ transplantation. Often associated with immunosuppressive therapy (tacrolimus or cyclosporine A), GVHD, or underlying opportunistic infections
Malignant HTN TMA	TMA precipitated by chronic, severe uncontrolled HTN. Acute but not chronic end-organ injury may improve with control of blood pressure
DIC	Coagulopathy with TMA caused by underlying condition, most often sepsis, malignancy, trauma, obstetric complications, or hematologic disorder
APLS	TMA in context of underlying autoimmune disease and meeting positive diagnostic criteria for APLS
Pregnancy-associated TMA (HELLP syndrome, preeclampsia)	TMA associated with obstetrical complications. Presence of significant proteinuria and de novo HTN are concerning for preeclampsia. Treatments can include control of BP and delivery

APLS, antiphospholipid syndrome; BP, blood pressure; CM-HUS, complement-mediated hemolytic-uremic syndrome; DIC, disseminated intravascular coagulation; DI-TMA, drug-induced thrombotic microangiopathy; GVHD, graft versus host disease; HELLP, hemolysis, elevated liver enzyme, and low platelet syndrome; HTN, hypertension; IA-HUS, infection-associated hemolytic-uremic syndrome; MAHA-T, microangiopathic hemolytic anemia with thrombocytopenia; TA-TMA, transplant-associated thrombotic microangiopathy; TMA, thrombotic microangiopathy; TTP, thrombotic thrombocytopenic purpura; VEGF, vascular endothelial growth factor.

## Data Availability

Not applicable.

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
