# Peer review of "Thrombotic Thrombocytopenic Purpura: Pathophysiology, Diagnosis, and Management"

_jcm, 2021, doi:10.3390/jcm10030536_

Round 1

Reviewer 1 Report

The authors illustrated comprehensively the state-of-art on thrombotic thrombocytopenic purpura (TTP). They addressed all the critical aspects associated with the disease, moving through the pathophysiology, diagnosis and therapies. I appreciated that the structure of the review was divided into sub-paragraphs, facilitating the understanding of the concepts listed.

Author Response

Journal of Clinical Medicine, Special Issue: The Latest Clinical Advances in Thrombocytopenia

Response to Reviewers:

We would like to express our gratitude to the Editors and Reviewers of this manuscript for carefully going through our manuscript and providing their thoughtful and highly constructive comments for improvement. We have revised the manuscript and made appropriate changes as detailed below and in the manuscript. All revisions are provided in red font in the manuscript document.  

Reviewer #1:

The authors illustrated comprehensively the state-of-art on thrombotic thrombocytopenic purpura (TTP). They addressed all the critical aspects associated with the disease, moving through the pathophysiology, diagnosis and therapies. I appreciated that the structure of the review was divided into sub-paragraphs, facilitating the understanding of the concepts listed.

We thank Reviewer one for her/his comments.

Reviewer #2:

Dr. Sukumar and colleagues have done a wonderful job addressing aspects of the pathophysiology, diagnosis and management of congenital and autoimmune TTP. Their review includes discussion of potential upcoming therapeutics in TTP and notes the importance of the multitude of significant sequalae that patients with TTP are at an increased risk for. For this they are to be commended. With this said, from a management aspect in the 2020 decade a review of TTP should note the importance of equitable and affordable access to the TTP therapeutic armamentarium. Specifically, there is no discussion of cost of therapy and one mention of quality-of-life near the conclusion (in reference to the sequalae mentioned prior, line 573). In addition to this, please find 4 major and 2 minor queries below. Thank you to the authors for writing this review.

We thank Reviewer two for her/his valuable comments. As suggested in the queries below a line about the cost-effectiveness of therapy has been highlighted.

Major

Line 23-26“Front line therapy includes daily plasma exchange with fresh frozen plasma replacement, corticosteroids, and if available, anti-VWF therapy with caplacizumab. Additional immunosuppression targeting ADAMTS13 autoantibodies includes the humanized anti-CD20 monoclonal antibody rituximab which is also frequently added to the initial therapy.”

In 2020 rituximab is an integral component of TTP therapy, not an additional immunosuppressant. As Coppo et al. most recently accurately captured (PMID 33150928), rituximab is part of front line therapy (as opposed to what it was historically: “salvage”). Indeed the authors appropriately detail this themselves on page 11 of 31 with regards to rituximab. The authors should note clearly that rituximab is part of front line therapy and can do so by going the Coppo et al. route and referring to frontline therapy in TTP as plasma exchange, immunosuppression with corticosteroids and rituximab, as well as caplacizumab.

 We agree with the reviewer that rituximab is useful in the front-line setting and for pre-emptive treatment, but recognize that there has not been a prospective randomized controlled trial or official registered indication of rituximab for this purpose. We have changed the text in the abstract to now read “Front line therapy includes daily plasma exchange with fresh frozen plasma replacement and immunosuppression with corticosteroids. Immunosuppression targeting ADAMTS13 autoantibodies with the humanized anti-CD20 monoclonal antibody rituximab is frequently added to the initial therapy. If available, anti-VWF therapy with caplacizumab is also added to the front line setting.” (Lines 23-27)

Line 252-254: As currently written I question the precision of this sentence “The pentad though was reported at a time long before the discovery of the effectiveness of plasma-based therapy to treat TTP.” If the authors are referring to the 25-year separation between the pentad being described in the Baltimore series (1966) and the Rock et al paper in the early 1990s, they may be bypassing Bukowski’s et al.’s mid/late 1970s publication in Blood (PMID 560229) describing the use of plasmapheresis in TTP (which itself was derived in simplifying whole blood exchange reported in the late 1950s by Rubinstein et al.).

 The sentence in question has been revised to “The pentad though was reported at a time before the effectiveness of plasma-based therapy to treat TTP was firmly established” (Line 252-254) so as to more accurately convey its meaning.

Line 442-458: As regards the caplacizumab part of the section entitled "Anti-VWF Strategy", the limitation of significant societal cost due to caplacizumab has now been reported and needs to be acknowledged, as noted in Goshua et al.'s publication in Blood (PMID 33280030) on the lack of cost effectiveness of caplacizumab.

 The following sentences addressing cost effectiveness ave been added and we cite Goshua et al.’s publication: “As a novel agent, one limitation of incorporating caplacizumab into current standard practice is its high cost. At its current price level (in the United States) as of 2020, a recent analysis suggested that the addition of caplacizumab to the front line treatment for all patients with iTTP would not be cost-effective [171]” (Lines 456-459)  

Line 467: The Seattle trial referenced is outdated. It is listed at 3 participants with a completion date in July 2017. If there is another trial please cite it and if not then would remove this sentence “A phase 1 trial evaluating the efficacy of NAC in iTTP is currently underway (https://www.clinicaltrials.gov/ct2/show/study/NCT01808521).”

We agree with the reviewer and this sentence has been removed.

Minor

Line 42: “dubbed” typically refers to nicknames or unofficial names, authors may consider replacing this word depending on what meaning they are going after.

Line 42 “dubbed” was changed to “named”

Line 95: The cited study included patients from 2000 through to end of 2015, inclusive. This said, I am not sure what “(Dec 2015)” refers to at the conclusion of the sentence.

(Dec 2015) was written at the end of the statement on prevalence (which is distinct from incidence) in the paper from which this line was referenced so was included to preserve accuracy of the text. For clarity, the end of the sentence now reads “as of December 2015.” (Line 95)

Reviewer #3

The paper "TTP: pathophysiology, diagnosis and management" is extremely well written and structured. The authors are labelled as “references” for this area. Figure 1 is very useful in understanding the molecular mechanism of the anomaly. The chapter on caplacizumab provides an original and very useful synthesis to this new review.

This outstanding manuscript can be published as it stands. It can be presumed that it will be read by many colleagues involved in the topic, both in biology and for the treatment of TTP.

We thank the reviewer for her/his kind comments.

Reviewer 2 Report

Dr. Sukumar and colleagues have done a wonderful job addressing aspects of the pathophysiology, diagnosis and management of congenital and autoimmune TTP. Their review includes discussion of potential upcoming therapeutics in TTP and notes the importance of the multitude of significant sequalae that patients with TTP are at an increased risk for. For this they are to be commended. With this said, from a management aspect in the 2020 decade a review of TTP should note the importance of equitable and affordable access to the TTP therapeutic armamentarium. Specifically, there is no discussion of cost of therapy and one mention of quality-of-life near the conclusion (in reference to the sequalae mentioned prior, line 573). In addition to this, please find 4 major and 2 minor queries below. Thank you to the authors for writing this review.

Major

Line 23-26: “Front line therapy includes daily plasma exchange with fresh frozen plasma replacement, corticosteroids, and if available, anti-VWF therapy with caplacizumab. Additional immunosuppression targeting ADAMTS13 autoantibodies includes the humanized anti-CD20 monoclonal antibody rituximab which is also frequently added to the initial therapy.”

In 2020 rituximab is an integral component of TTP therapy, not an additional immunosuppressant. As Coppo et al. most recently accurately captured (PMID 33150928), rituximab is part of front line therapy (as opposed to what it was historically: “salvage”). Indeed the authors appropriately detail this themselves on page 11 of 31 with regards to rituximab. The authors should note clearly that rituximab is part of front line therapy and can do so by going the Coppo et al. route and referring to frontline therapy in TTP as plasma exchange, immunosuppression with corticosteroids and rituximab, as well as caplacizumab.

Line 252-254: As currently written I question the precision of this sentence “The pentad though was reported at a time long before the discovery of the effectiveness of plasma-based therapy to treat TTP.” If the authors are referring to the 25-year separation between the pentad being described in the Baltimore series (1966) and the Rock et al paper in the early 1990s, they may be bypassing Bukowski’s et al.’s mid/late 1970s publication in Blood (PMID 560229) describing the use of plasmapheresis in TTP (which itself was derived in simplifying whole blood exchange reported in the late 1950s by Rubinstein et al.).

Line 442-458: As regards the caplacizumab part of the section entitled "Anti-VWF Strategy", the limitation of significant societal cost due to caplacizumab has now been reported and needs to be acknowledged, as noted in Goshua et al.'s publication in Blood (PMID 33280030) on the lack of cost effectiveness of caplacizumab.

Line 467: The Seattle trial referenced is outdated. It is listed at 3 participants with a completion date in July 2017. If there is another trial please cite it and if not then would remove this sentence “A phase 1 trial evaluating the efficacy of NAC in iTTP is currently underway (https://www.clinicaltrials.gov/ct2/show/study/NCT01808521).”

Minor

Line 42: “dubbed” typically refers to nicknames or unofficial names, authors may consider replacing this word depending on what meaning they are going after.

Line 95: The cited study included patients from 2000 through to end of 2015, inclusive. This said, I am not sure what “(Dec 2015)” refers to at the conclusion of the sentence.

Author Response

(The authors gave the same response as above.)

Reviewer 3 Report

The paper "TTP: pathophysiology, diagnosis and management" is extremely well written and structured. The authors are labelled as “references” for this area. Figure 1 is very useful in understanding the molecular mechanism of the anomaly. The chapter on caplacizumab provides an original and very useful synthesis to this new review.

This outstanding manuscript can be published as it stands. It can be presumed that it will be read by many colleagues involved in the topic, both in biology and for the treatment of TTP.

Author Response

(The authors gave the same response as above.)

Round 2

Reviewer 2 Report

Dr. Sukumar and colleagues have addressed the 4 major and 2 minor queries. Thank you for writing this review and congratulations on an addition to the literature that will be cited many times.